# The Role of PAR2 in MASLD Progression and HCC Development

**DOI:** 10.3390/ijms26157076

**Published:** 2025-07-23

**Authors:** Pietro Guerra, Patrizia Pontisso, Andrea Martini

**Affiliations:** 1Department of Medicine, University of Padova, Via Giustiniani 2, 35128 Padova, Italy; pietro.guerra1@gmail.com; 2Azienda Ospedale-Università Padova, Via Giustiniani 2, 35128 Padova, Italy; andrea.martini@aopd.veneto.it

**Keywords:** metabolic dysfunction, liver cancer, protease-activated receptor

## Abstract

Metabolic dysfunction-associated steatotic liver disease (MASLD) has recently become the leading cause of chronic liver disease and can progress to hepatocellular carcinoma (HCC) through multiple pathogenic mechanisms. Protease-activated receptor 2 (PAR2) is a G-protein-coupled receptor activated by proteases such as trypsin, tryptase or coagulation factors VII and Xa. Recent studies have shown that PAR2 expression is increased in the liver of patients with MASLD or liver fibrosis. Its activation is linked to metabolic dysfunction through several pathways, including SREBP1c activation, AMPK inhibition and Akt-induced insulin resistance. Inhibition of PAR2 has been effective in reducing MASLD progression in different animal models. Notably, PAR2 blockade has also been effective in more advanced stages of the disease by dampening chronic inflammation and fibrogenesis through the inhibition of hepatic stellate cell activation and of TGF-β and SerpinB3 production. PAR2 also plays a role in cancer development, promoting tumour proliferation, angiogenesis and expression of immune checkpoint inhibitors (like PD-L1, CD47 and CD24). Due to its multifaceted involvement in liver disease, PAR2 is emerging as a key therapeutic target in this clinical context. This review aims to summarise current knowledge on PAR2′s role in MASLD and its potential as a therapeutic target.

## 1. Introduction

Protease-activated receptor 2 (PAR2) is a member of the G-protein-coupled receptor (GPCR) family. Unlike classical GPCRs that are activated by ligand binding, PAR2 and other members of the PAR family are uniquely activated through proteolytic cleavage of their N-terminal region by specific serine proteases. This cleavage unveils a tethered ligand, which interacts with the receptor itself to trigger a conformational change and initiate intracellular signalling. Among the four known PARs, PAR2 is selectively activated by various enzymes, including trypsin, tryptase and coagulation factors VIIa and Xa, among others [1].

PAR2 plays a crucial role in inflammation and fibrosis deposition and has recently been identified as a key regulator of metabolism. Consequently, it has been extensively studied in the context of obesity, liver disease progression and cancer development (Figure 1). In this review, we explore the pathophysiological roles of PAR2 in chronic liver disease progression and its potential as a therapeutic target.

## 2. Metabolic Dysfunction—Associated Liver Disease and Liver Cancer

Metabolic dysfunction-associated steatotic liver disease (MASLD), formerly known as non-alcoholic fatty liver disease (NAFLD), has recently emerged as the leading cause of chronic liver disease, with an estimated global prevalence of 30% [2]. It encompasses a spectrum of progressive stages, beginning with steatotic liver disease (SLD)—characterised by lipid accumulation in the hepatocytes—and advancing to inflammation and metabolic dysfunction-associated steatohepatitis (MASH), previously referred to as non-alcoholic steatohepatitis (NASH). The inflammatory process promotes fibrotic tissue deposition, which may progress to liver cirrhosis and, ultimately, hepatocellular carcinoma (HCC) [2] (Figure 1). Due to its central role in the pathogenesis of liver cirrhosis, MASLD has been extensively investigated as a driver of liver-related complications. Moreover, growing evidence links MASLD to cardiovascular disease and extrahepatic malignancies [3], which are now recognised as the leading causes of death among these patients [4,5].

MASLD progression to HCC does not depend on cirrhotic evolution, as it is a recognised risk factor even in the absence of liver cirrhosis [6]. It has been estimated that in Italy, nearly 50% of patients with HCC in the context of metabolic liver disease are not cirrhotic [7]. There are different risk factors for cancer development, including obesity, metabolic dysfunction, type 2 diabetes, metabolic syndrome and insulin resistance [8].

MASLD can progress to HCC through different pathogenetic mechanisms. In brief, metabolic dysfunction, which may occur in genetically predisposed individuals, leads to an imbalance in lipid metabolism, resulting in increased oxidative stress and DNA damage. This process is accompanied by chronic inflammation, which impairs immune surveillance and promotes fibrosis deposition and HCC progression [9,10]. Obesity is a driver in MASLD progression to cancer, since it is associated with systemic inflammation and insulin resistance. These factors lead to deterioration in the regulation of immune surveillance and lipid metabolism in the liver. In fact, obesity alone doubles the risk of developing HCC, and this risk quadruples in individuals with a body mass index (BMI) exceeding 35 kg/m^2^ [6].

The following sections review how PAR2 activation contributes to the onset and progression of metabolic dysfunction, inflammation, fibrosis and ultimately, to cancer development.

## 3. PAR2 and Metabolism

Given the central role of metabolic dysfunction in the onset of MASLD, increasing attention has been directed toward molecular mediators that link these processes. Among them, PAR2 has emerged as a key player in metabolic regulation (Figure 1) and different studies have proposed it as a possible therapeutic target (Table 1).

### 3.1. Glucose Metabolism and Insulin Resistance

Insulin resistance, a hallmark of type 2 diabetes and obesity, is central in the pathogenesis of metabolic dysfunction and hepatic steatosis. Impaired insulin signalling leads to a reduction in protein kinase B (Akt) activation, exacerbating dysregulation of glucose and lipid metabolism. A recent study investigated the role of PAR2 in promoting insulin resistance [11]. The authors observed an increased expression of PAR2 in hepatocytes from patients with concurrent diabetes and hepatic steatosis. Using a murine model of steatosis and diabetes, they also demonstrated that genetic deletion of PAR2 (PAR2-KO) improved histological markers of steatosis activity and led to a reduction in plasma glucose and insulin levels. These effects were primarily mediated by an increased expression of the glucose transporter type 2 (GLUT2), which permits glucose uptake and glycogen storage in the hepatocytes. Mechanistically, they showed that PAR2 activation inhibited insulin-Akt signalling, promoting insulin resistance, whereas pharmacological inhibition or genetic silencing of PAR2 restored insulin sensitivity.

The contribution of PAR2 to insulin resistance was also reported in another study where the authors found elevated levels of forkhead transcription factor (FoxO) 6 in insulin-resistant rats [12]. FoxO6 upregulation was accompanied by increased PAR2 expression. Deletion of FoxO6 improved insulin sensitivity and reduced PAR2 expression. The mechanistic link between FoxO6 and PAR2 was found in IL-1β, a cytokine, induced by FoxO6 signalling, which in turn stimulated PAR2 expression and inhibited insulin signalling.

### 3.2. Lipid Metabolism and Obesity

Insulin resistance promotes lipid uptake, de novo lipogenesis and lipid storage, ultimately leading to hepatic steatosis. Lipid metabolism represents another critical pathway in the pathogenesis of MASLD; therefore, the role of PAR2 was also evaluated in lipid homeostasis. A recent study observed that hepatic PAR2 expression was elevated in steatotic livers and that patients with high PAR2 levels in the liver had significantly increased plasma LDL cholesterol [13]. In PAR2-KO mice fed a high-fat diet (HFD), PAR2 deficiency was associated with lower hepatic and plasma cholesterol levels. This effect was mediated by three mechanisms: (a) reduced cholesterol synthesis, (b) increased hepatic cholesterol uptake and (c) enhanced biliary cholesterol excretion, as demonstrated by PCR analysis of gene transcription. Despite unchanged plasma triglyceride levels, PAR2-KO mice exhibited reduced hepatic accumulation of triglyceride and fatty acid. This was accompanied by downregulation of key lipogenic enzymes and upregulation of genes associated with β-oxidation. Mechanistically, the authors proved that PAR2-induced activation of JNK1/2 promoted sterol regulatory element binding protein 1 (SREBP1c) activation and inhibited AMP-activated protein kinase (AMPK), a key regulator in lipid catabolism.

Further evidence supporting the link between PAR2 and AMPK came from Kim et al., who demonstrated that PAR2-KO mice were protected from developing hepatic steatosis when fed on an HFD. PAR2-KO mice showed higher AMPK activation compared to wild-type controls. Conversely, in vitro, overexpression of PAR2 was associated with AMPK inhibition. The downregulation of AMPK by PAR2 resulted in an impairment of autophagy, contributing to hepatic lipid accumulation [14].

Badeanlou et al. reported that genetic deletion of PAR2 or tissue factor (TF), which mediates coagulation factor VIIa-induced PAR2 activation, protected mice from developing obesity and insulin resistance when subjected to an HFD [15]. Furthermore, PAR2 deletion was associated with a reduction in macrophage infiltration in the adipose tissue, supporting the role of PAR2 in the development of tissue inflammation. Interestingly, the researchers uncovered a dual role for PAR2. Specific deletion of PAR2 or TF in non-hematopoietic cells led to a reduction in weight gain. Conversely, deletion of these targets in immune cells did not affect obesity development but led to decreased markers of insulin resistance and inflammation.

### 3.3. Liver Steatosis and Pathology

Some years later, the same group explored the role of TF–PAR2 signalling in the liver [16]. Genetic deletion or pharmacological inhibition of TF, PAR2 or both led to reductions in gluconeogenesis, lipogenesis, and liver inflammation in a murine model of diet-induced obesity. Notably, these beneficial effects were also observed when TF or PAR2 deletion was restricted to hematopoietic cells, further supporting the role of this receptor in the crosstalk between the immune system and target tissue. Interestingly, wild-type (WT) mice fed an HFD exhibited increased hepatic infiltration of CD11b^+^CD11c^+^ macrophages and CD8^+^ lymphocytes, compared to mice fed a low-fat diet (LFD). Accordingly, TF deletion decreased CD11b^+^CD11c^+^ macrophage infiltration, while PAR2 deletion reduced the CD8^+^ T cell population.

A novel mechanism involving PAR2 in liver disease was recently identified by Villano et al., who demonstrated an interaction between PAR2 and SerpinB3 [17]. PAR2 activation induced the expression of the transcription factor CCAAT/enhancer-binding protein β (C/EBPβ), a known regulator of SerpinB3. In turn, SerpinB3, a member of the serine protease inhibitor family, was shown to be essential for PAR2 activation. Importantly, the authors found that PAR2 activity could be effectively inhibited using a small molecule called 1-Piperidine Propionic Acid (1-PPA): treatment with 1-PPA prevented lipid accumulation, inflammation and liver fibrosis in mice overexpressing SerpinB3. Additionally, it suppressed C/EBPβ expression in both THP-1 and HepG2 cells.

SerpinB3 has been previously implicated in the pathogenesis of steatotic liver disease. In a recent study, due to its involvement in lipid accumulation and inflammation, it was proposed as a new *hepatokine* [18]. In this study, two mouse models were used: one overexpressing SerpinB3 (TG/SB3) and the other expressing an isoform lacking its antiprotease activity (KO/SB3). Mice were fed on two different steatogenic diets: methionine–choline-deficient (MCD) or choline-deficient, L-amino acid-defined (CDAA). Overexpression of SerpinB3 resulted in increased hepatic lipid accumulation and inflammation, while loss of its activity conferred protection compared to WT mice. Moreover, TG/SB3 mice exhibited increased hepatic accumulation of crown-like structures formed by macrophages, a characteristic histological feature of steatotic liver disease, mirroring the findings of Wang et al. regarding TF–PAR2 signalling and CD11b^+^CD11c^+^ macrophage recruitment [16].

Lipid accumulation, when excessive, can lead to hepatotoxicity. This process, together with insulin resistance and obesity, is often accompanied by the development of chronic inflammation.

## 4. PAR2 and Inflammation

Inflammation is a process that allows the organism to cope with external damage. However, its dysregulation can be the trigger of various diseases. In the context of liver disease, problems begin when the inflammatory response becomes chronic. This drives the progression from benign MASLD to more advanced forms, such as MASH. PAR2 has been associated with several signalling pathways that contribute to the onset of inflammation in the liver (Figure 1) and other organs. Complementing these findings, various animal models have suggested it as a potential therapeutic target (Table 2).

### 4.1. Lung and Airways

Recent evidence has highlighted its involvement in promoting antigen responses during allergic reactions. A recent study demonstrates that inhibition of PAR2 reduces the production of inflammatory cytokines in a model of airway hypersensitivity [32]. It has also been shown that PAR2 antagonism effectively reduces airway hyperresponsiveness and inflammation in a murine model of allergic reaction [19]. Similar results were obtained in a mouse model of allergen-induced asthma [20], and in vitro [33].

### 4.2. Sepsis and Infection

In line with these findings, our research team recently published a study evaluating the efficacy of PAR2 antagonism using 1-PPA in a murine model of sepsis induced by intraperitoneal injection of lipopolysaccharide [21]. Treatment with 1-PPA significantly attenuated the inflammatory response and vasodilation, consequently enhancing cardiac function, reducing organ damage and alleviating clinical symptoms, which ultimately led to improved survival rates. Similar results were previously obtained in a model of liver damage induced by LPS administration. The authors reported an increased expression of PAR2 in Kupffer cells, tissue resident macrophages involved in pathogen-response and immune-activation or tolerance. Notably, PAR2 inhibition resulted in a suppression of TNFα (tumour necrosis factor) and an improved healing process [22].

The role of PAR2 in the response to microorganisms was also highlighted by Chu et al., who described a novel PAR2-dependent mechanism of neutrophil activation, driving inflammation and giving the rationale for inhibiting this pathway in the treatment of neutrophil-mediated inflammatory disease [34]. Other evidence demonstrated that PAR2 inhibition can prevent infection by *Candida albicans* [35] and periodontal inflammation induced by *Porphyromonas gingivalis* [23].

### 4.3. Neuroinflammation

Our research team recently published findings on the effects of PAR2 inhibition using 1-PPA in neuroinflammation, demonstrating its efficacy in reducing amyloid deposition and neuroglial inflammation in fibroblasts in patients with Parkinson’s disease [36]. Consistently, it has been shown that inhibition of mast cell tryptase effectively reduced PAR2-driven neuroinflammation in a murine model of cardiac arrest [24].

### 4.4. Gut and Microbiome

Several studies have also explored the role of PAR2 in intestinal diseases. It has been reported that a microbiome with high proteolytic activity can trigger colitis via PAR2 activation in mice [25]. Bacterial proteolytic activation of PAR2 has also been identified as a potential therapeutic target in inflammatory bowel disease [26] and in mediating both inflammation and pain in colitis [37]. However, Ke et al. showed that PAR2 deficiency in myeloid-derived suppressor cells enhanced their immunosuppressive activity, suggesting a dual role for this receptor in this context [27].

### 4.5. Kidney and Vascular System

Of particular relevance to this review is the link between PAR2, inflammation and metabolism, an interplay that involves not only the liver but also other organs. Ha et al. showed that PAR2-deficient mice were protected from HFD-induced kidney inflammation and injury [28]. In a subsequent study, the same group demonstrated that PAR2-KO mice were also protected from age-related kidney inflammation and senescence [29].

Two different studies further demonstrated the efficacy of PAR2 inhibition in preventing LDL-induced vascular damage in models of atherosclerosis, reinforcing the role of this receptor in the crosstalk between metabolism and inflammation [38,39].

### 4.6. PAR2 Dual Role in Inflammation

More recent studies have provided further insights into the complex interaction between the immune system and target organs (Figure 2). Reches et al. addressed this issue by investigating the role of PAR2 in two distinct murine models of liver injury: one immune-mediated (using concanavalin A) and one toxin-induced (with carbon tetrachloride, CCL_4_). The authors compared wild-type and PAR2 knock-out mice and repeated the experiments after bone marrow transplantation between the two strains. Their results demonstrated that PAR2 expression in liver tissue was essential for hepatocyte regeneration after toxic injury, whereas PAR2 expression in immune cells enhanced inflammation and worsened immune-mediated damage [30].

This dual role of PAR2 was later confirmed by the same group in a model of autoimmune diabetes. In this context, PAR2 expression in pancreatic β-cells was protective against immune damage, while its expression in lymphocytes promoted β-cell destruction and diabetes onset [31].

This dichotomous role of PAR2 in immune and target tissues warrants further investigation, particularly given the emerging importance in tumour development of this receptor.

## 5. PAR2 and Fibrosis

Chronic inflammation ultimately leads to fibrotic tissue deposition. Currently, the only therapeutic approach capable of halting—or, in certain cases, reversing—fibrosis relies on the removal or control of the underlying etiological factor. Owing to its critical role in the development of liver-related complications, fibrosis remains a major focus of translational research. Figure 1 outlines the mechanistic involvement of PAR2 in the initiation and perpetuation of fibrogenesis, while Table 3 summarises key preclinical studies investigating PAR2 inhibition as a potential antifibrotic strategy.

### 5.1. Liver Fibrosis

Inhibition of PAR2 using the pepducin PZ-235 was reported as effective in preventing liver fibrosis in a two-murine model, one fed on an HFD and the other treated with CCL_4_. PZ-235 directly reduced hepatic stellate cell (HSC) activation and decreased hepatocyte death, thereby limiting the main stimulus for chronic inflammation [40].

Other studies have confirmed the involvement of PAR2 in modulating HSC function. Knight et al. showed that PAR2-KO mice were protected from fibrosis induced by CCl_4_, a protection associated with reduced expression of TGF-β, the main mediator of fibrogenesis [41]. In a later study, the same authors investigated the relationship between TF activation and liver fibrosis using three different genetically modified murine models lacking PAR2, the cytoplasmic domain of TF, or both. Deletion of either PAR2 or TF conferred protection against fibrosis, and simultaneous deletion of both did not produce any additive benefit, suggesting that TF-PAR2 signalling is a shared pathway driving fibrosis [42].

In vitro studies also support this evidence. HSCs upregulated both PAR1 and PAR2 expression when cultured on plastic to promote myofibroblastic phenotype transformation. Stimulation of either receptor increased proliferation and activation of mitogen-activated protein kinase (MAPK), while their inhibition had the opposite effect [49]. Another in vitro study explored the role of PAR2 in apoptosis. Activation-induced cell death, triggered by phorbol myristate acetate, was suppressed by concurrent PAR2 stimulation with tryptase, suggesting a protective anti-apoptotic role of PAR2 in activated HSCs [50].

### 5.2. Other Organs Fibrosis

The pro-fibrotic activity of PAR2 is not limited to the liver. Tisch et al. recently reported that genetic deletion of PAR2 reduced airway fibrosis following allergen exposure [43]. Similar protective effects were observed in the kidney, where PAR2KO mice were protected from renal inflammation and fibrosis [44]. Along the same lines, both PAR2 and PAR1 contribute to kidney injury and fibrosis in a diabetic mouse model [45].

Additional studies have examined the effects of PAR2 on intestinal fibrosis. Mast cell-derived tryptase activated PAR2 on intestinal fibroblasts, promoting their transition to myofibroblasts and mirroring the findings in hepatic HSCs [46]. Two additional studies confirmed that PAR2 inhibition protected against intestinal fibrosis [47,48].

Collectively, these findings underscore the role of PAR2 in orchestrating immune responses and promoting fibrosis across multiple organ systems [51]. This has important implications, as sustained inflammation and fibrotic signalling may contribute to tumour development via activation of oncogenic pathways.

## 6. PAR2 and Cancer

PAR2 expression and activation has also been associated with intracellular signalling pathways associated with cancer development and immune-evasion (Figure 1 and Figure 2). Table 4 summarises studies investigating the role of PAR2 in this context and its inhibition as a potential therapeutic target.

### 6.1. Cancer Progression and Growth

In the liver, PAR2 expression in HSCs promotes HCC growth and angiogenesis in a murine xenograft model, whereas genetic deletion of PAR2 suppresses these effects. Furthermore, PAR2-deficient HSCs exhibited reduced activation following TGF-β stimulation and Hep3B-conditioned medium, indicating that PAR2 expression in HSCs contributes not only to fibrogenesis but also to HCC progression [52].

Moreover, a recent study showed that high PAR2 expression in HCC tissue samples after radical resection was associated with more aggressive clinical features. Patients with elevated PAR2 levels had larger tumours, more advanced stages and a higher incidence of microvascular invasion. Building on these findings, the authors demonstrated in two in vitro hepatic cancer cell models that PAR2 overexpression enhanced both proliferation and metastatic potential, while silencing of PAR2 reversed these effects. In vivo, PAR2 silencing reduced the extent of metastatic spread in a mouse model of liver metastasis [53].

In another study, the authors showed that secreted cathepsin S interacts with PAR2 to regulate the transition of cancer stem cells in HCC. Cathepsin S was secreted by CD47^+^ cells displaying stemness characteristics. Suppression of CD47 reduced HCC cell proliferation, and disruption of the cathepsin S/PAR2 axis increased chemosensitivity, suggesting a role for this pathway in therapy resistance [54].

### 6.2. Immune Surveillance and Immunotherapy

Immune checkpoint inhibitors (ICIs) have become the first-line therapy for various cancers. Given its role in modulating immune responses, PAR2 has been investigated in this context. In a recent study, a novel gene signature predictive of response to immunotherapy across multiple cancer types was identified [59]. Interestingly, among the genes analysed, PAR2, together with RNA-binding motif protein 9, was associated with poorer response to ICI therapy and T cell dysfunction.

The role of PAR2 in regulating anti-tumour immune responses was further explored in a study on metastatic colorectal cancer [55]. The authors found that PAR2 expression correlated with reduced macrophage-mediated phagocytosis of malignant cells. This effect was driven by increased expression of the “don’t eat me” signal, CD24. Conversely, neutrophil elastase, which cleaves and inhibits PAR2, downregulates CD24 expression, thereby enhancing phagocytosis. Similar findings were also reported in breast cancer, where activation of PAR2 signalling via coagulation factor VIIa led to reduced phagocytosis in an in vitro model [56]. Moreover, genetic deletion of PAR2 in vivo resulted in a smaller tumour size. Both studies highlighted a synergistic effect between PAR2 inhibition (whether pharmacological or genetic) and ICI treatment.

Apart from phagocytosis, other authors have focused on the effects of PAR2 in regulating T-cell response. A recent study reported that activation of PAR2 by coagulation factor Xa (FXa) conferred resistance to anoikis in HCC cells, thereby promoting metastasis. FXa-mediated PAR2 activation also increased PD-L1 expression while simultaneously inhibiting CD8^+^ T cell infiltration into tumours. Inhibition of FXa in vivo resulted in decreased metastasis and downregulation of PD-L1. These effects were further enhanced by concomitant administration of anti-PD-L1 antibodies [57]. Similar results were reported in the context of breast cancer, where PAR2 activation leads to increased production and stabilisation of PD-L1 and a reduction in CD8^+^ T cell activity [58].

These findings suggest that PAR2 expression in cancer cells not only promotes tumour progression and invasion but also suppresses the immune response. On the other hand, as previously reported, PAR2 expression in the immune system may activate an immune response (Figure 2). However, to the best of our knowledge, no study has directly investigated the impact of PAR2 activation on immune cells within the tumour microenvironment.

## 7. Discussion and Future Perspectives

### 7.1. PAR2 as a Therapeutic Target in Liver Disease Progression

The role of PAR2 in promoting metabolic dysfunction has been extensively investigated. Reducing insulin resistance may represent a promising therapeutic strategy for obesity, diabetes and MASLD, as it constitutes one of the key pathogenetic drivers of these conditions [60]. Several preclinical studies have demonstrated the beneficial effects of PAR2 inhibition, with no significant side effects or adverse reactions reported (Table 1). Moreover, these findings are consistent with the observed upregulation of PAR2 expression in liver biopsies from patients with diabetes, MASLD or fibrosis [11,13].

As lipid accumulation progresses, it can lead to hepatocellular injury and ultimately to chronic inflammation, resulting in fibrotic tissue deposition. There is growing evidence supporting the efficacy of PAR2 inhibitors in attenuating liver fibrosis (Table 3). Notably, the antifibrotic effects of PAR2 inhibition are not solely attributable to the reduction in hepatic steatosis but also involve the direct modulation of chronic inflammation and fibrogenesis, particularly through the suppression of hepatic stellate cell activation and TGF-β production. Indeed, PAR2 inhibition has been effective in reducing fibrosis both in dietary preclinical models of hepatic steatosis (e.g., HFD or CDAA diets) and in toxin-induced liver injury models (e.g., CCl_4_) (Table 3). It is important to note that, among PAR2 antagonists, PZ-235 pepducin (Oasis Pharmaceuticals) is currently undergoing clinical development for various indications. This compound has advanced to phase 1 clinical trials for non-alcoholic steatohepatitis (NCT05680233). Meanwhile, the clinical development process for 1-PPA is still in progress.

### 7.2. PAR2 as a Therapeutic Target in HCC

A critical consequence of chronic liver inflammation is the increased risk of tumour development. The incidence of HCC is known to rise not only in the advanced stages of cirrhosis but also in the setting of MASH. Several mechanisms contribute to this elevated oncogenic risk, including enhanced DNA damage, persistent proliferative signalling and impaired immune surveillance [2].

Taken together, the aforementioned findings highlight the fundamentally pro-inflammatory role of PAR2. Non-tissue-specific inhibition of PAR2 has been shown to exert beneficial effects in various conditions characterised by uncontrolled inflammation (Table 2). PAR2 blockade has demonstrated efficacy in preclinical models of asthma, autoimmune diseases and septic shock, among others.

The dual role of PAR2—stimulating the immune system when expressed by immune cells and suppressing immune responses when expressed by target tissue cells—seems to function as a balance that shifts depending on the clinical context. Indeed, the observation that systemic inhibition of PAR2 in pro-inflammatory settings confers therapeutic benefit suggests that, in these conditions, immune activation is dysregulated and that PAR2 inhibition contributes to the restoration of immune homeostasis. However, under specific conditions of acute tissue injury, PAR2 activation has been associated with increased survival of organ-specific cells and protection against tissue damage and cell death. These findings imply that PAR2 may exert context-dependent protective effects in the acute or chronic setting.

Extensive evidence has investigated the role of PAR2 in cancer, demonstrating that its activation promotes cancer proliferation, migratory capacity and invasiveness of tumour cells (Figure 2). On the same line, PAR2 inhibition has been shown to reduce the risk of metastasis as well as the growth of various tumour types, including HCC. Additionally, PAR2 contributes to immune escape mechanisms when expressed in the tumoural cells, consistent with its ability to upregulate different immune checkpoint inhibitors (Figure 2). On the other hand, PAR2 appears to enhance inflammation in other clinical contexts, and studies reported an increased immune response when it was activated in immune system cells. Given that current oncological strategies increasingly focus on modulating the immune system, it is important to highlight the potential contribution of PAR2 in this context. However, studies on PAR2’s potential activation within the immune-system cells are lacking. The current available evidence suggests that inhibiting PAR2 in tumoural cells enhances therapeutic response.

### 7.3. PAR2 as a Prognostic Marker in HCC

Given its broad biological functions, PAR2 expression could be investigated as a potential prognostic biomarker in liver disease and cancer. However, current evidence in this field remains limited.

Rana et al. [13] and Shearer et al. [11] have shown that PAR2 is overexpressed in liver biopsies from patients with metabolic liver disease, with expression levels increasing alongside the severity of fibrosis and inflammation. These findings suggest a correlation between PAR2 expression and disease progression, although direct prognostic implications have not been fully explored.

In this context, different molecules should also be considered as surrogate markers or readouts of PAR2 activation. For example, SerpinB3 is known to induce PAR2 synthesis and activation and has been described as a prognostic factor in various liver conditions. Numerous studies have supported SerpinB3 association with HCC development [61,62] and worse outcomes in patients with liver cirrhosis [63,64], cholangiocarcinoma [65,66] and HCC [67,68]. Also, Li et al. [57] have reported that the prognosis of patients with HCC is influenced by the expression of coagulation factor Xa, a known activator of PAR2. However, these studies did not directly investigate PAR2 expression in hepatic tissue, highlighting a significant gap in our understanding of the mechanistic link and underscoring the need for further research in this area.

To the best of our knowledge, only two studies have directly associated PAR2 expression with clinical outcomes in HCC. Tsai et al. found that high PAR2 expression correlates with features of more aggressive tumours at diagnosis and poorer prognosis in HCC patients after liver resection [69], while Barenche et al. reported that PAR2 expression is associated with reduced response to immunotherapy in different types of cancer [59].

Nevertheless, these findings require confirmation through larger, well-designed observational clinical studies to validate the prognostic role of PAR2.

## 8. Conclusions

In conclusion, recent research has thoroughly examined the crucial role of PAR2 in the progression of metabolic dysfunction-associated steatotic liver disease (MASLD). PAR2 acts as an upstream regulator in various pathogenetic mechanisms associated with MASLD. Consequently, inhibiting PAR2 emerges as a promising therapeutic strategy, potentially interrupting simultaneously multiple disease-driving pathways (Figure 3). However, clinical validation through prospective studies is warranted to confirm these promising preclinical findings and to establish the efficacy and safety of PAR2 inhibition in humans.

Increasing evidence also points to a key role for PAR2 in promoting tumour growth and immune escape. Notably, PAR2 acts upstream of multiple signalling pathways whose downstream effectors are currently being targeted by tailored therapies (Figure 3). Unlike these single-pathway inhibitors, targeting PAR2 may allow for a broader therapeutic effect, intercepting several pro-inflammatory, pro-fibrogenic and pro-tumourigenic signals, as already described. Nevertheless, PAR2′s role in immuno-oncology remains underexplored, presenting a significant opportunity for further investigation. This emerging area of research is particularly promising, given the expanding role of immunotherapy in cancer treatment and the potential for PAR2 to modulate immune responses within the tumour microenvironment.

## 9. Patents

The authors declare that P.P. and M.A. are listed as inventors of the following patents: IT 102017000026858, EP 20709715.5, US 11628163, IT 102019000012930, EP 20709715.5, US 17/629.791 pending; P.P. is listed as inventor of the patents IT 102022000014593, PTC/IB2023/057138 pending. All the patents have been filed by the University of Padova.

## Figures and Tables

**Figure 1 ijms-26-07076-f001:**
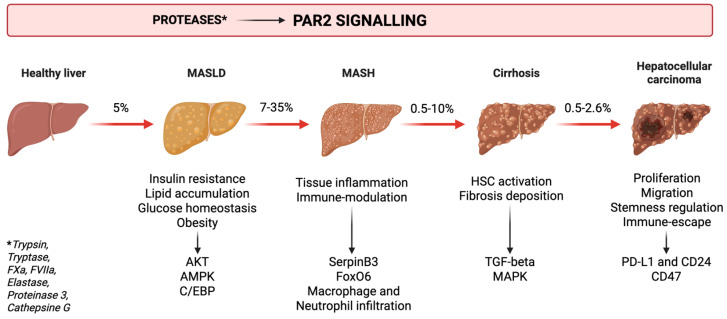
Protease-activated receptor 2 (PAR2) signalling role in the development and progression of metabolic dysfunction-associated liver disease (MASLD). The figure illustrates the sequential stages of chronic liver disease from MASLD to metabolic dysfunction-associated steatohepatitis (MASH), cirrhosis and hepatocellular carcinoma (HCC), along with the estimated annual progression rates between each stage. This schematic illustration depicts the role of PAR2 in both the establishment and progression of each stage of liver disease, highlighting the intracellular signalling pathways driven by PAR2 activation.

**Figure 2 ijms-26-07076-f002:**
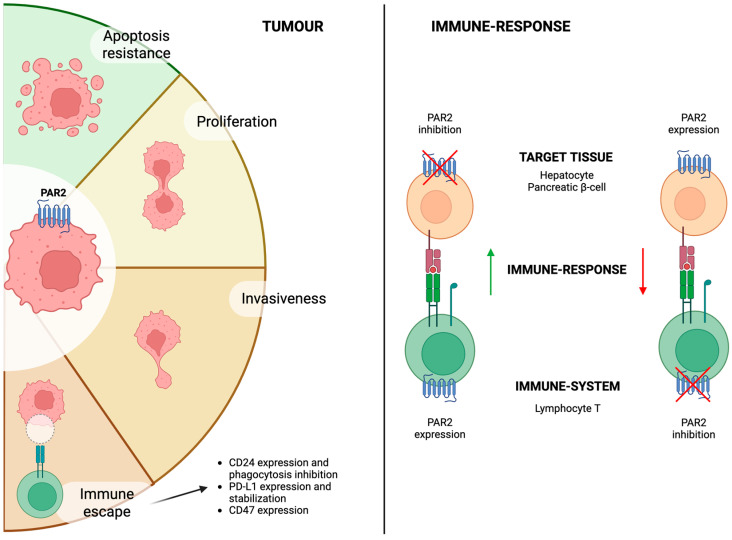
The dichotomous roles of PAR2 in cancer biology. Left panel: PAR2 expression in cancer cells promotes tumour progression by enhancing apoptosis resistance, proliferation, invasiveness, and ultimately facilitating immune escape through the upregulation of immune checkpoint inhibitors. Right panel: PAR2 expression in target cells—such as hepatocytes and pancreatic β-cells—can promote immune tolerance. Conversely, its expression in T lymphocytes can stimulate immune response.

**Figure 3 ijms-26-07076-f003:**
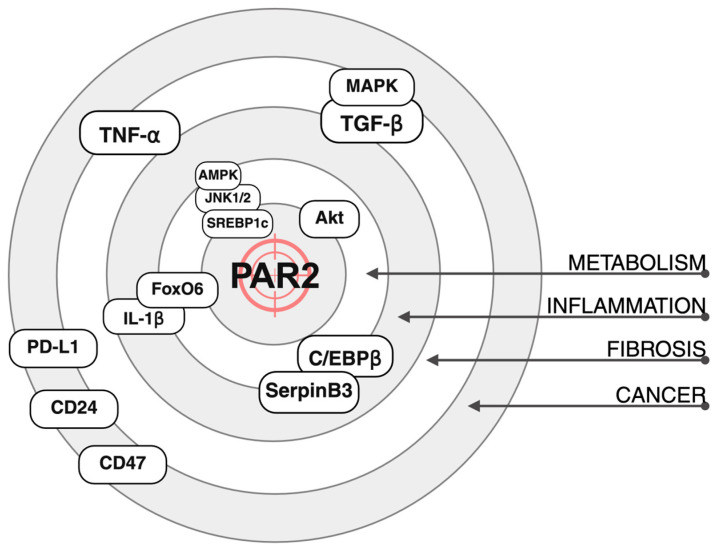
Protease-activated receptor 2 (PAR2) as a therapeutic target for different pathogenetic mechanisms. The figure schematically illustrates the primary signalling pathways affected by PAR2 inhibition, highlighting their contributions to the dysregulation of metabolism, inflammation, fibrosis and cancer. Akt, Protein Kinase B. AMPK, AMP-Activated Protein Kinase. C/EBP-β, CCAAT/Enhancer-Binding Protein β. FoxO6, Forkhead Transcription Factor 6. MAPK, Mitogen-Activated Protein Kinase. SERNPB1c, Sterol Regulatory Element Binding Protein 1. TNF-α, Tumour Necrosis Factor α. TGF-β, transforming growth factor β.

**Table 1 ijms-26-07076-t001:** Summary of preclinical studies targeting protease-activated receptor 2 (PAR2) in animal models of metabolic dysfunction. HFD, high-fat diet. KO, knock-out. 1-PPA, 1-Piperidine propionic acid. MCD, methionine–choline deficient. CDAA, choline–deficient, l-amino acid-defined.

Reference	Mouse Strain	Diet/Model	PAR2 Inhibition	Outcome
**METABOLISM**				
Shearer et al., 2022 [11]	C57BL/6J C57BL/6 Db/Db	HFD, HFD + streptozotocin	Whole-body KO PZ-235 (PAR2-inhibitor)	Reduced hepatic steatosis. Increased glucose metabolism.
Kim et al., 2019 [12]	C57BL/6J	Standard diet	Whole-body KO	Reduced hepatic inflammation and
Rana et al., 2019 [13]	C57BL/6J	HFD	Whole-body KO	hepatic steatosis. Increased lipid metabolism.
Kim et al., 2021 [14]	C57BL/6J	HFD	Whole-body KO	Reduced hepatic steatosis.
Badeanlou et al., 2011 [15]	C57BL/6J	HFD	Whole-body KO Bone-marrow chimeras	Reduced obesity development and adipose tissue inflammation, hepatic steatosis and inflammation.
Wang et al., 2015 [16]	C57BL/6J	HFD	Whole-body KO Bone-marrow chimeras	Reduced insulin resistance and obesity development.
Villano et al., 2024 [17]	C57BL/6J, BALB/c	MCD and CDAA	1-PPA (PAR2-inhibitor)	Reduced hepatic steatosis, inflammation and fibrosis.

**Table 2 ijms-26-07076-t002:** Summary of preclinical studies targeting protease-activated receptor 2 (PAR2) in animal models of inflammation. LPS, lipopolysaccharides. AOM, Azoxymethane. DSS, Dextran Sulfate Sodium. HFD, high-fat diet. CCL_4_, Carbon Tetrachloride. KO, knock-out. 1-PPA, 1-Piperidine propionic acid.

Reference	Mouse/Rat Strain	Diet/Model	PAR2 Inhibition	Measured Outcomes
**INFLAMMATION**				
Schiff et al., 2023 [19]	BALB/c	Allergen sensitisation and challenge	C781 (PAR2 inhibitor)	Reduced airway hyperresponsiveness.
De Matos et al., 2022 [20]	BALB/c	Allergic-ovalbumin lung inflammation	ENMD1068 (PAR2 inhibitor)	Reduced airway inflammation.
Luisetto et al., 2024 [21]	C57BL/6J	LPS injection	1-PPA (PAR2 inhibitor)	Reduced organ damage and mortality. Improved cardiac function.
Jesmin et al., 2006 [22]	Wistar rats	LPS injection	PAR2 blocking peptide (PAR2 BP)	Increased tissue healing and decreased fibrosis deposition.
Francis et al., 2023 [23]	BALB/c	P. gingivalis-induced periodontitis	GB88 (PAR2 inhibitor)	Reduced bone loss and inflammation.
Ocak et al., 2020 [24]	Sprague–Dawley rats	Asphyxia-induced cardiac arrest	APC366 (mast cell tryptase inhibitor)	Reduced neurological deficit and neuroinflammation.
Santiago et al., 2023 [25]	C57BL/6N, Nod2−/−	Crohn’s disease colonising microbiota	R38E-PAR2 (PAR2 cleavage-resistant mouse)	Reduced colitis development and severity.
Rondeau et al., 2024 [26]	C57BL/6N	Proteolytic bacteria colonisation	R38E-PAR2 (PAR2 cleavage-resistant mouse)	Reduced colitis development and severity.
Ke et al., 2020 [27]	C57BL/6	AOM/DSS colitis-associated cancer	Whole-body KO	Increased cancer development.
Ha et al., 2022 [28]	C57BL/6J	HFD	Whole-body KO	Improved kidney function.
Ha et al., 2024 [29]	C57BL/6J	Renal fibrosis (adenine diet or cisplatin injection)	Whole-body KO	Reduced renal fibrosis and inflammation.
Reches et al., 2022 [30]	C57BL/6J	Concanavalin A CCl_4_	Whole-body KO Bone-marrow chimeras	Reduced hepatic damage.
Reches et al., 2024 [31]	C57BL/6J/NOD	Autoimmune diabetes	Tissue-specific PAR2 deletion	Reduced pancreatic damage and T1D onset.

**Table 3 ijms-26-07076-t003:** Summary of preclinical studies targeting PAR2 in animal models of fibrosis. HFD, high-fat diet. KO, knock-out. 1-PPA, 1-Piperidine propionic acid. CCL4, Carbon Tetrachloride. MCD, methionine–choline deficient. HDM, house dust mite. NOS, endothelial nitric oxide synthase. TNBS, 2,4,6-Trinitrobenzene Sulfonic Acid.

Reference	Mouse Strain	Diet/Model	PAR2 Inhibition	Measured Outcomes
**FIBROSIS**				
Shearer et al., 2016 [40]	C57BL/6	MCD CCl_4_	PZ-235 (PAR2 inhibitor)	Reduced liver fibrosis.
Knight et al., 2012 [41]	C57BL/6	CCl_4_	Whole-body KO	Reduced liver fibrosis.
Knight et al., 2017 [42]	C57BL/6	CCl_4_	Whole-body KO	Reduced liver fibrosis.
Tisch et al., 2025 [43]	C57BL/6	HDM	Whole-body KO	Reduced lung fibrosis.
Ha et al., 2022 [44]	Wild-type mice	Adenine diet	Whole-body KO	Reduced kidney fibrosis.
Mitsui et al., 2020 [45]	Akita mice (eNOS-/-)	Diabetic kidney disease	FSLLRY-NH2 (PAR2 inhibitor)	Reduced kidney fibrosis.
Liu et al., 2021 [46]	C57BL/6	DSS-induced colitis model	ENMD-1068 (PAR2 inhibitor)	Reduced intestinal fibrosis.
Liu et al., 2024 [47]	C57BL/6	TNBS-induced colitis	Whole-body KO	Reduced intestinal fibrosis.
Xie et al., 2022 [48]	SAMP1/YitFc, 129Sv/J, CD-1	Salmonella-infected mice and TNBS	Elafin (Cathepsin-S inhibitor)	Reduced intestinal fibrosis.

**Table 4 ijms-26-07076-t004:** Summary of preclinical studies targeting PAR2 in animal models of cancer. HSC, hepatic stellate cell. LPS, lipopolysaccharide.

Reference	Mouse Strain	Diet/Model	PAR2 Inhibition	Measured Outcomes
**CANCER**				
Mußbach et al., 2016 [52]	SCID	Xenotransplantation (HSC and LX-2 cells)	shRNA silencing	Reduced tumour growth and angiogenesis.
Sun et al., 2018 [53]	BALB/c nude	Xenotransplantation (HepG2 and SMMC-7721)	shRNA silencing	Reduced tumour growth and metastasis.
Lee et al., 2014 [54]	BALB/c nude	Xenograft (PCL/PRF/5 cells)	shRNA silencing CD47	Reduced chemoresistance and tumour growth.
Liu et al., 2025 [55]	BALB/c nude	LPS injection Xenograft (HT-29 cells)	shRNA silencing, GB88 (PAR2 inhibitor)	Enhanced phagocytosis (downregulation of CD24).
Ghosh et al., 2025 [56]	BALB/c	Allograft model (4T1 cells)	shRNA silencing	Enhanced phagocytosis (downregulation of CD47).
Li et al., 2024 [57]	NOD/SCID, C57BL/6	Xenograft (MHCC97-H and Hepal-6 cells))	FXa inhibition	Reduced immune-escape (downregulation of PD-L1).
Paul et al., 2023 [58]	BALB/c	Allograft model (4T1 cells)	shRNA silencing	Reduced immune-escape (downregulation of PD-L1).

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
