# Peer review of "The Role of PAR2 in MASLD Progression and HCC Development"

_ijms, 2025, doi:10.3390/ijms26157076_

Round 1

Reviewer 1 Report

Comments and Suggestions for Authors

This review informs us that protease-activated receptor 2 (PAR2), a member of the G protein-coupled receptor (GPCR) family, is activated by proteolytic cleavage by serine proteases such as trypsin, factor VIIa, and elastase. Its activation triggers various intracellular signaling pathways involved in inflammation, cell proliferation, angiogenesis, and migration, key processes in tumor progression.
The review is interesting, but I believe changes should be made to the text and presentation that would improve the review.
(1) The abstract should clearly describe the function of the PAR2 receptor; it is ambiguous and simplistic.
(2) Lines 68 referring to PAR2 AND METABOLISM, line 153 PAR2 AND INFLAMMATION, and line 210 PAR2 AND FIBROSIS should include figures summarizing the PAR2 mechanisms involved in the presentation of inflammation and fibrosis.
(3) Tabular summaries should also be provided, representing a sequence of all studies related to PAR2 and its relationship to liver physiology and pathophysiology.
(4) The relationship between PAR2 and Kupffer cells, which are important in liver and immune disorders, should be discussed.
2
(5) The conclusion contains very basic information on the relationship between PAR2 and liver inflammation, as well as its possible relationship with liver cancer. I suggest a conclusion more related to the mechanisms involved in the development of liver tumors.
(6) I recommend developing a methodology for selecting bibliographic references, e.g., inclusion/exclusion criteria.

Reviewer 2 Report

Comments and Suggestions for Authors

This review article provides an extensive and up-to-date synthesis of the literature on Protease-Activated Receptor 2 (PAR2), emphasizing its role in metabolic dysfunction-associated steatotic liver disease (MASLD), fibrosis, inflammation, and hepatocellular carcinoma (HCC). The manuscript is comprehensive, well-structured, and clearly written, demonstrating the authors' strong grasp of the topic.

Major Comments

  1. The manuscript synthesizes a vast body of recent and relevant literature across metabolism, immunology, fibrosis, and oncology. While the breadth is impressive, the depth of mechanistic insight in each section is inconsistent. Some mechanisms (e.g., TF–PAR2 axis) are well detailed, while others (e.g., immune-TME crosstalk) are only superficially addressed. A critical appraisal of gaps in knowledge is missing. Include a section or table summarizing current knowledge gaps and future research directions.
  2. The structure is largely topic-based (e.g., “PAR2 and metabolism,” “PAR2 and fibrosis”) but lacks transitional flow. The narrative sometimes feels fragmented. Consider including a “bridging paragraph” at the start or end of each section to link themes (e.g., connecting metabolic regulation and immune modulation).
  3. Figure 1 and Figure 2 are referenced but not included in the provided document or not clearly visible for review. These figures are crucial for visualizing complex pathways discussed. Ensure all referenced figures are included, high-quality, and effectively summarize key signaling and pathophysiological mechanisms.
  4. The manuscript occasionally reads like a collection of findings without sufficient critical evaluation (e.g., whether findings from PAR2 knockout models have clinical correlates, or how contradictory findings are resolved).Add brief critical reflections or commentary after key studies, particularly where translational relevance or study limitations exist.
  5. The dichotomy in PAR2’s role (pro-tumor in hepatocytes vs. immunostimulatory in immune cells) is intriguing but underdeveloped. Expand this theme, perhaps as a dedicated section. Discuss how tissue-specific modulation or targeted delivery could exploit this dual role therapeutically.
  6. The manuscript discusses pharmacological inhibition (e.g., 1-PPA, pepducins) in multiple sections, but the therapeutic landscape is not synthesized or compared. Add a concluding section or table summarizing known PAR2 inhibitors, their targets, study models, and current development stage (preclinical/clinical).
  7. In-text citations are extensive, but some key original studies are buried under review citations, and some references (e.g., Refs 10, 17, 23) are cited multiple times without adding novel insights each time. Prioritize original research citations and avoid redundancy.

Minor Comments

Section

Comment

Abstract

The term “knockouts” is used inappropriately to describe reduced protein levels; this should be corrected.

Introduction

Clearly distinguish MASLD and MASH terminology early. Consider mentioning the global burden.

Page 3

Consider defining key abbreviations (e.g., GLUT2, FoxO6) at first mention for clarity.

Typographical

Repeated minor grammatical issues (e.g., "in among MASLD patients", "Conversely, PAR2 expression...").

Section headers

Use consistent capitalization and style (e.g., "PAR2 and Metabolism" vs. "PAR2 AND METABOLISM").

Conclusion

The conclusion is appropriate but could be stronger by including specific recommendations or a graphical summary of PAR2’s multi-system role.

Author Contributions

Incomplete. Please add funding source and conflict of interest declaration as per journal requirements.

Reviewer 3 Report

Comments and Suggestions for Authors

Thank you for submitting your manuscript. This review article provides a comprehensive and timely synthesis of current literature regarding the role of Protease-Activated Receptor 2 (PAR2) in the progression from Metabolic Dysfunction-Associated Steatotic Liver Disease (MASLD) to Hepatocellular Carcinoma (HCC). The manuscript is well-structured, extensively referenced, and includes illustrative figures that enhance comprehension. However, to elevate the manuscript to publication standards, I suggest addressing the following comments related to scientific clarity, completeness, critical interpretation, and formatting

Comments

  1. The current title is informative but verbose. Consider a more concise version to improve impact, such as: “The Role of PAR2 in MASLD Progression and HCC Development”.
  2. The abstract is descriptive but lacks critical synthesis. Please include specific data trends or quantitative summaries from the cited literature (e.g., percentage reduction in inflammation/fibrosis upon PAR2 inhibition).
  3. The manuscript is primarily a summary of existing literature. It would benefit from critical evaluation of conflicting results. For instance: In what contexts is PAR2 protective versus pathogenic? Are findings from murine models consistent with human clinical observations?
  4. The dual role of PAR2 in both promoting and suppressing inflammation depending on cellular context (e.g., hepatocytes vs. immune cells) is mentioned but not fully reconciled. Please discuss: How might this duality influence therapeutic targeting? Could cell-specific delivery of inhibitors be a feasible strategy?
  5. A dedicated “Limitations and Future Perspectives” section would improve completeness. For example: What are the barriers to translating PAR2-targeting agents (e.g., 1-PPA, GB88) to clinical settings? What gaps exist in understanding PAR2's role in human MASLD or HCC?
  6. Consider including a summary table listing:
  • Key studies on PAR2
  • Disease model (e.g., MASH, cirrhosis, cancer)
  • Intervention used (e.g., genetic deletion, pharmacological inhibition)
  • Major outcome
  1. Figure 1 is informative. However: Please clarify the timeline of MASLD stages shown. Indicate which enzymes activate PAR2 (e.g., trypsin, factor Xa).
  2. Figure 2 effectively presents PAR2’s dichotomous role. Suggest adding clear icon legends and labelling immune cells (e.g., CD8⁺ T cells, macrophages). Please explain how "immune escape" is mediated mechanistically.
  3. There is inconsistent terminology regarding liver disease (e.g., “MASLD” vs. “NAFLD”). Since the field is transitioning nomenclature, a brief note explaining the equivalence or difference would be useful for readers.
  4. Avoid repetition and wordy constructions. Please revise placeholder lines: Line 312: “Please add: ‘This review received no external funding.’” Line 314: “Declare conflicts of interest…” → should be finalized.
  5. The language is generally readable, but multiple grammatical inconsistencies and awkward constructions are present.
  6. Minor grammatical issues persist: Line 300: "On the other hand, conversely..." — revise to one conjunction. Line 65: “...including cirrhosis and hepatocellular carcinoma. .” → remove duplicate period.
  7. The manuscript follows standard review format well (abstract → body → conclusion → references), but lacks section numbering or subheadings within major sections. Consider dividing long sections (e.g., inflammation, fibrosis) into subsections for better readability.
  8. Abbreviations are explained at the end (line 316), but first mentions in the main text should also be expanded (e.g., MASH, AMPK, FoxO6, etc.).
  9. Please ensure all figures are labeled correctly and consistently referenced in the text. Figure captions could include more explanatory content (e.g., what is depicted in each quadrant of the tumor illustration in figure 2).
  10. Please ensure uniform formatting of references: Some references lack journal volume or issue details (e.g., #9, #52). Verify DOI consistency and correct line wrapping. Remove any residual placeholder-style citations (e.g., “Firstname Lastname” in the editorial line).
  11. The conclusion (lines 302–308) is succinct, but it could benefit from more specific calls for future research, e.g., “clinical validation of PAR2 inhibitors,” “immune cell-specific PAR2 knockout models,” etc.

Specific Questions for Authors

1: Did any study investigate PAR2 expression in liver biopsies of MASLD patients with vs. without HCC? If not, would this be a promising area for future biomarker work?

2: could PAR2 serve as a diagnostic or prognostic marker (e.g., high expression correlating with tumor aggressiveness or therapy response)?

3: Are there any known off-target effects of PAR2 inhibitors like 1-PPA or GB88 in preclinical models?

4: Given that PAR2 modulates PD-L1 expression, how does its role compare with other immune-modulatory GPCRs (e.g., PAR1, CCR5)?

5: Could you elaborate on whether human clinical trials or tissue studies support PAR2's translational relevance? If not, highlight this as a gap.

6: Would PAR2 inhibitors be expected to work systemically or need targeted delivery? Have there been toxicity assessments?

7: how does PAR2 expression change over time in MASLD progression? Could it be used for disease staging or risk prediction?

8: Can you provide a table summarizing animal models used in the key studies cited?

Suggestions for authors

  1. Add a table summarizing major findings from key studies (model system, PAR2 manipulation, observed effects on steatosis, fibrosis, HCC) would aid clarity.
  2. Include a Future Directions Section, a short paragraph or boxed text summarizing open research questions and directions for translational development.
  3. Consider elaborating more on the “double-edged sword” nature of PAR2—pro-tumor in some contexts, yet immune-enhancing in others. This may have major therapeutic implications.
  4. Adding a third schematic that maps PAR2 roles across MASLD progression (healthy → steatosis → fibrosis → HCC) would unify the manuscript.
  5. Discuss Biomarker Potential. Could PAR2 serve as a diagnostic or prognostic biomarker in MASLD or HCC? Any early human data to support this?

Round 2

Reviewer 1 Report

Comments and Suggestions for Authors

Dear Authors

The responses to the suggestions and comments have been implemented correctly. There has been a substantial improvement in the review, which includes tables that adequately summarise the previous studies for the present review. Therefore, I consider that the present review should be accepted in its current form.

Reviewer 2 Report

Comments and Suggestions for Authors

The authors have responded to all the comments successfully.